# The Impact of the Virulence of *Pseudomonas aeruginosa* Isolated from Dogs

**DOI:** 10.3390/vetsci10050343

**Published:** 2023-05-11

**Authors:** Telma de Sousa, Andreia Garcês, Augusto Silva, Ricardo Lopes, Nuno Alegria, Michel Hébraud, Gilberto Igrejas, Patricia Poeta

**Affiliations:** 1Department of Genetics and Biotechnology, University of Trás-os-Montes and Alto Douro (UTAD), 5000-801 Vila Real, Portugal; 2Microbiology and Antibiotic Resistance Team (MicroART), Department of Veterinary Sciences, University of Trás-os-Montes and Alto Douro (UTAD), 5000-801 Vila Real, Portugal; 3Functional Genomics and Proteomics Unit, University of Trás-os-Montes and Alto Douro (UTAD), 5000-801 Vila Real, Portugal; 4Associate Laboratory for Green Chemistry (LAQV), Chemistry Department, Faculty of Science and Technology, University Nova of Lisbon, 2829-516 Lisbon, Portugal; 5CRL-CESPU, Cooperativa de Ensino Superior Politécnico e Universitário, R. Central Dada Gandra, 1317, 4585-116 Gandra, Portugal; 6CITAB, University of Trás-os-Montes and Alto Douro (UTAD), 5000-801 Vila Real, Portugal; 7INNO—Veterinary Laboratory, R. Cândido de Sousa 15, 4710-503 Braga, Portugal; 8Department of Veterinary Sciences, School of Agrarian and Veterinary Sciences, University of Trás-os-Montes and Alto Douro (UTAD), 5000-801 Vila Real, Portugal; 9UMR Microbiologie Environnement Digestif Santé (MEDiS), INRAE, Université Clermont Auvergne, 60122 Saint-Genès-Champanelle, France; 10Veterinary and Animal Research Centre (CECAV), University of Trás-os-Montes and Alto Douro (UTAD), 5000-801 Vila Real, Portugal; 11Associate Laboratory for Animal and Veterinary Sciences (AL4AnimalS), University of Trás-os-Montes and Alto Douro (UTAD), 5000-801 Vila Real, Portugal

**Keywords:** *Pseudomonas aeruginosa*, dogs, antibiotic resistance

## Abstract

**Simple Summary:**

This study aimed to evaluate the antimicrobial resistance patterns and biofilm production of clinical isolates of *Pseudomonas aeruginosa* which is a pathogenic bacterium that can cause infections in dogs. The results showed widespread resistance to various β-lactam antimicrobials, with amikacin and tobramycin being the only effective aminoglycosides. All isolates carried the *opr*D gene, which is essential in governing the entry of antibiotics into bacterial cells, and all isolates also carried virulence genes. The study emphasizes the importance of continued monitoring of antimicrobial resistance in veterinary medicine and responsible antibiotic use to prevent multi-drug resistance from emerging. The findings of this study have significant implications for the treatment and prevention of *P. aeruginosa* infections in dogs and highlight the need for further research to better understand the mechanisms underlying the emergence of multi-drug resistance.

**Abstract:**

*Pseudomonas aeruginosa* is a pathogenic bacterium that can cause serious infections in both humans and animals, including dogs. Treatment of this bacterium is challenging because some strains have developed multi-drug resistance. This study aimed to evaluate the antimicrobial resistance patterns and biofilm production of clinical isolates of *P. aeruginosa* obtained from dogs. The study found that resistance to various β-lactam antimicrobials was widespread, with cefovecin and ceftiofur showing resistance in 74% and 59% of the isolates tested, respectively. Among the aminoglycosides, all strains showed susceptibility to amikacin and tobramycin, while gentamicin resistance was observed in 7% of the tested isolates. Furthermore, all isolates carried the *opr*D gene, which is essential in governing the entry of antibiotics into bacterial cells. The study also investigated the presence of virulence genes and found that all isolates carried *exo*S, *exo*A, *exo*T, *exo*Y, *apr*A, *alg*D, and *plc*H genes. This study compared *P. aeruginosa* resistance patterns worldwide, emphasizing regional understanding and responsible antibiotic use to prevent multi-drug resistance from emerging. In general, the results of this study emphasize the importance of the continued monitoring of antimicrobial resistance in veterinary medicine.

## 1. Introduction

*Pseudomonas aeruginosa* is a Gram-negative bacterium that is known to cause infections in humans, particularly in individuals with weakened immune systems or those with underlying medical conditions such as cystic fibrosis. This bacterium is highly adaptable and is capable of surviving in a wide range of environments, including in water, soil, and other moist environments [1]. It is also resistant to many antibiotics, making it difficult to treat. *P. aeruginosa* is known to excrete several virulence factors, including toxins and enzymes that contribute to tissue damage and disease [2].

One of the most relevant virulence factors of this pathogen is its ability to form biofilms, complex communities of bacteria encased in a protective matrix of extracellular polymeric substances (EPS). Biofilms can adhere to several surfaces like medical devices or lung tissue, for example, making it difficult for the host immune system or antibiotics to eliminate the infection [3,4]. In addition, the EPS matrix of the biofilm can facilitate the exchange of genetic material between bacteria, allowing for the spread of antibiotic resistance genes. The formation of a biofilm by *P. aeruginosa* involves several stages, including initial attachment, microcolony formation, EPS production, and maturation [5]. The EPS matrix is composed of a variety of polysaccharides, proteins, and extracellular DNA, helping to provide structural support and protection for the bacteria within the biofilm [6]. Therefore, strategies to prevent or disrupt biofilm formation in *P. aeruginosa* are important for public health, particularly in healthcare settings [7].

*Pseudomonas* also excrete a variety of toxins that contribute to the virulence of this bacterium [8]. These include exotoxin A, which inhibits protein synthesis, and pyocyanin, which generates reactive oxygen species that can damage host cells. The bacterium can also excrete elastase, which degrades host tissues and interferes with the immune response [9,10,11]. Finally, *P. aeruginosa* has several mechanisms for evading the host immune system, such as producing pigments that make it difficult for the immune cells to and modifying the bacterium’s lipopolysaccharide structure in order to evade recognition by the immune system [12].

*Pseudomonas* can cause infections in dogs, particularly those with weakened immune systems or underlying medical conditions. Dogs can acquire *P. aeruginosa* infections through a variety of means, such as contact with contaminated water, soil, or surfaces [13]. The process of an infection in dogs typically involves four stages: exposure, colonization, invasion, and dissemination. Exposure occurs when a dog comes into contact with a pathogen, such as through contact with contaminated surfaces, inhalation of infected droplets, or bites from infected animals. Colonization follows, during which the pathogen multiplies and takes up residence in a specific area of the dog’s body, such as the skin, respiratory tract, or gastrointestinal tract. During this stage, the dog may not show any signs of infection. The third stage, invasion, occurs when the pathogen invades the dog’s tissues and causes inflammation, swelling, and the production of pus or other fluids. Finally, in the dissemination stage, the pathogen may spread from the initial site of infection to other parts of the body, including the bloodstream or organs. This can lead to a severe, life-threatening infection if left untreated [14]. The bacteria can cause a range of symptoms in dogs, including skin and ear infections, urinary tract infections, and respiratory infections. The treatment of *P. aeruginosa* is challenging because some strains have developed multidrug resistance [15]. The bacterium’s high intrinsic antibiotic resistance is caused by several factors, including low outer membrane permeability, the production and derepression of the chromosomal AmpC β-lactamase, and the presence of numerous genes coding for multidrug resistance efflux pumps [16,17,18]. *P. aeruginosa* is typically resistant to many antibiotics, including penicillins, first- and second-generation cephalosporins, macrolides, chloramphenicol, and some aminoglycosides (such as streptomycin, neomycin, kanamycin, and spectinomycin), tetracyclines, and sulfonamides [16,19,20]. However, some antibiotics can be effective in treating *P. aeruginosa* infections, such as ureidopenicillins, carboxypenicillins, third- and fourth-generation cephalosporins, carbapenems, aminoglycosides, fluoroquinolones, and polymyxins [21,22,23].

Studying the relationship between *P. aeruginosa* and dogs is important for several reasons. Understanding the mechanisms of resistance can help veterinarians to choose the most effective treatment options for dogs with *P. aeruginosa* infections [24,25]. Another point is that dogs can act as reservoirs of *P. aeruginosa*, potentially spreading the bacteria to other animals or humans. Thus, understanding the transmission and prevention of *P. aeruginosa* infections in dogs can have implications for public health, as has been suggested by others [26]. Finally, *P. aeruginosa* is an important model organism for studying bacterial pathogenesis and antibiotic resistance mechanisms [27]. By studying *P. aeruginosa* infections in dogs, researchers can gain insights into the broader mechanisms of bacterial infections and resistance, which can lead to the development of new treatments and preventative measures [1]. The objective of this research was to concentrate on the *P. aeruginosa* that was obtained from samples of dogs at the INNO Veterinary Laboratory. The aim was to analyze the phenotype and genotype of antimicrobial resistance.

## 2. Materials and Methods

### 2.1. Samples and Bacterial Isolates

During the period from November 2021 to December 2021, a total of 27 *P. aeruginosa* isolates were obtained from various pathologies at the INNO Veterinary Laboratory (Table 1). INNO is the leading reference laboratory in providing specialized services for veterinary medicine in Braga, Portugal, and all isolates in this work originate from different areas of the country. The identification of each strain was confirmed using VITEK 2^®^ COMPACT (bioMérieux, Marcy-l′Étoile, France); additionally, they were seeded on a *Pseudomonas* agar base supplemented with a CN (Liofilchem, Roseto Degli, Abruzzi, Italy) medium at 37 °C for 24–48 h in the medical microbiology laboratory. The isolates were subsequently cryopreserved at −20 °C in skim milk.

### 2.2. Antimicrobial Susceptibility Testing

The antimicrobial susceptibilities for all *P. aeruginosa* isolates were determined using th4e Kirby–Bauer disk diffusion method in accordance with EUCAST standards (2022). A total of 11 antibiotics were utilized in this study, including ceftazidime (CAZ, 30 μg/disk), cefepime (FEP, 30 μg/disk), doripenem (DOR, 10 μg/disk), imipenem (IMI, 10 μg/disk), meropenem (MEM, 10 μg/disk), aztreonam (ATM, 30 μg/disk), tobramycin (TOB, 10 μg/disk), ciprofloxacin (CIP, 5 μg/disk), gentamicin (CN, 10 μg/disk), amikacin (AK, 30 μg/disk) and ticarcillin-clavulanic acid (TTC, 85 μg/disk). The selection of these antibiotics and their corresponding cut-off values were based on EUCAST 2022 (EUCAST, 2022), with the exception of ceftazidime, which was based on CLSI 2021 (CLSI, 2021) due to the differing concentration of the disk from EUCAST standards. The antibiotics enrofloxacin (ENR), marbofloxacin (MAR), ceftiofur (CEFT), and cefovecin (CEF) were assessed by VITEK 2® COMPACT (bioMérieux).

### 2.3. Biofilm Formation and Biomass Quantification

The bacterial adhesion of all isolates was evaluated using a microtitre plate-based assay with modifications, as previously described [28]. To perform the assay, one colony from each bacterial culture that had grown overnight on brain–heart infusion (BHI) agar was suspended in Luria–Bertani (LB) broth and incubated for 24 h at 37 °C. Then, the bacterial suspension was diluted 0.5 on the McFarland scale using tryptic soy broth (TSB). Next, 100 µL of each bacterial suspension was added to eight wells of a flat-bottomed polystyrene microtitre plate, and the plate was incubated at 37 °C for 24 h. The negative control was sterile TSB, while the positive control was *Pseudomonas aeruginosa* ATCC 27853^®^ with the ability to form biofilm. After incubation, the plate was washed two times with distilled water and allowed to dry at room temperature. Then, 100 µL of crystal violet (CV) at 0.1% (*v*/*v*) was added to each well for 10–15 min. The excess stain was removed by washing the plate two times with distilled water, and then the plate was left to dry for several hours or overnight. For qualitative assays, wells were photographed when dry. To quantify the biofilm biomass, 100 µL of 30% (*v*/*v*) acetic acid was added to solubilize the CV, and the optical density was measured at 570 nm using a blank of uninoculated 30% acetic acid and a microplate reader (BioTek ELx808U, BioTek, Winooski, VT, USA).

### 2.4. DNA Extraction

The method used was the boil method [29]. Briefly, in order to extract DNA, two to three colonies from each *P. aeruginosa* isolate were collected and suspended in 500 µL of sterile distilled water. The suspension was vortexed vigorously and then subjected to a heat bath at 100 °C for 8 min. The samples were centrifuged for 2 min at 12,000 rpm and the pellets were then discarded. The total DNA concentration was determined using a NanoDrop system. The DNA concentration was calculated for each sample and subsequently adjusted to 200 μg/mL. The DFS-Taq DNA polymerase from Bioron was used. This has a range of 10–500 μg/mL.

### 2.5. Antimicrobial Resistance and Virulence Genes

Antimicrobial resistance genes were screened in all isolates based on their phenotypic resistance results. The genomic DNA of all bacterial samples were utilized as templates for the PCR amplification of the 16S rDNA gene, which was subsequently used to confirm the presence of *P. aeruginosa*. The two primers used were 27F (5′ AGAGTTTGATCCTGGCTCAG-3′) and 1495R (5′ CTACGGCTACCTTGTTACGA-3′). These functions as forward primer and reverse primer, respectively [30]. Based on their phenotypic resistance profile, each isolate underwent PCR screening for the presence of the following antimicrobial resistance genes: *bla*_TEM_, *bla*_SHV_, *bla*_CTX_, *bla*_PER_, *bla*_SME_, *bla*_KPC_, *bla*_IMP_ *bla*_Smp_, *bla*_Vim_, *bla*_Vim-2_, *bla*_NDM_, *bla*_OXA_, *aac*(6′)-Ie-*aph*(2″)-Ia, *aph*(3′)-IIIa, aac(3)-I, aac(3)-II, aac(3)-III, aac(3)-IV, *ant*(4′)-Ia and *ant*(2′)-Ia. All isolates were screened for genes encoding virulence factors by PCR: *pil*B, *pil*A, *apr*A, *tox*A, *tss*C, *plc*H, *las*A, *las*B, *las*R, *lasI*, *exo*U, *exo*S, *exo*A, *exo*Y, *exo*T, *rhl*R, *rhl*I, *rhl*A/B and *alg*D. The primer sequences for all genes are presented in Table 2. Positive controls were established for each gene using multiple strains from the molecular genetics’ laboratory, while Mili Q water was employed as a negative control. We utilized conventional PCR (single PCR) to detect resistance genes in accordance with the PCR cycles provided in the respective references in Table 2. On the other hand, for the detection of virulence genes, we primarily employed conventional PCR (single PCR) with the exception of the *exo*A, *exo*Y, *exo*T gene trio, and the *rh*lR and *rh*lI genes; these were identified using multiplex PCR, with the PCR cycles specified in the corresponding references.

## 3. Results and Discussion

Several antimicrobial agents used in both human and veterinary medicine were tested for their effectiveness against 27 clinical isolates of *P. aeruginosa*. While our investigation demonstrated that none of the isolates were multidrug-resistant to the tested antibiotics (i.e., resistance to three or more antibiotic classes), it is important to note that many other antibiotic classes remain to be examined. It is possible that some of these untested classes may have multidrug resistance in these isolates. This is an important finding as it suggests that the isolates that were analyzed may still be susceptible to certain antibiotics and can be treated effectively with the appropriate use of therapy. Resistance to various β-lactam antimicrobials was observed among the isolates tested. In particular, cefovecin showed resistance in 74% of the isolates and 59% of the isolates were resistant to ceftiofur (Table 3). Ceftiofur and cefovecin are both antibiotics that belong to the third-generation cephalosporin class commonly used in veterinary medicine for the treatment of bacterial infections, including in otitis in dogs in Portugal. Ceftiofur is usually administered parenterally, while cefovecin is available as an injectable long-acting formulation, with a single dose providing protection for up to 14 days. As with any antibiotic, the inappropriate use or overuse of ceftiofur and cefovecin in veterinary medicine can lead to the emergence and spread of antibiotic-resistant bacteria, including those causing otitis in dogs. Moreover, resistance to these antibiotics may already exist in bacterial populations due to their use in livestock farming. Therefore, it is essential to use these antibiotics judiciously and only when necessary on the basis of the results of diagnostic tests and under the guidance of a qualified veterinarian [51]. Imipenem demonstrated resistance in 30% of the isolates, while 26% of the isolates were resistant to meropenem. In contrast, only 4% of the isolates demonstrated resistance to ticarcillin-clavulanic acid. All isolates showed intermediate susceptibility to doripenem. Among the aminoglycosides, all strains showed susceptibility to amikacin and tobramycin. However, gentamicin showed 7% resistance in the tested isolates. The susceptibility patterns exhibited significant variation overall. Out of the three fluoroquinolone antibiotics examined, only 7% of the isolates demonstrated resistance to ciprofloxacin and enrofloxacin, whereas marbofloxacin showed resistance in only 4% of the isolates. A study conducted by Harada et al. investigated antimicrobial susceptibility and resistance mechanisms to anti-pseudomonal agents in *P. aeruginosa* isolates collected from dogs and cats in Japan [52]. A total of 73 *P. aeruginosa* isolates were collected and tested for resistance against six different antimicrobial agents: orbifloxacin, enrofloxacin, ciprofloxacin, cefotaxime, aztreonam, and gentamicin. The study found that the resistance rates against orbifloxacin, enrofloxacin, ciprofloxacin, cefotaxime, aztreonam, and gentamicin were 34.2%, 31.5%, 20.5%, 17.8%, 12.3%, and 4.1%, respectively. This study is the first report on cephalosporin- and fluoroquinolone-resistant isolates of *P. aeruginosa* from Japanese companion animals. The findings highlight the importance of surveillance of antimicrobial resistance in veterinary medicine and the need for appropriate antimicrobial use. Another study by Shahini et al. investigated the resistance patterns of *P. aeruginosa* strains isolated from different regions of Iran. For example, in Tehran, the highest levels of resistance were observed for trimethoprim (100%) and ceftazidime (80%), while imipenem (60%) and cefepime (52%) had the lowest resistance. Indifferent states of America, *Pseudomonas* showed the least resistance to imipenem (15%) and ciprofloxacin (20%), whereas gentamicin (50%) showed the highest resistance [53]. The findings mentioned in the statement suggest that different populations in different countries may have different resistance patterns. This can be influenced by various factors such as the usage of different antibiotics and hygiene standards. For instance, the high resistance levels observed in Tehran may be attributed to the frequent use of certain antibiotics in that region or to the poor hygiene standards found in healthcare facilities. On the other hand, the low resistance levels observed in America may be due to better adherence to infection control measures and the judicious use of antibiotics [5,53]. Overall, our study and others highlight the importance of understanding *P. aeruginosa* resistance patterns in different populations and regions. It also emphasizes the need for responsible use of antibiotics and strict adherence to infection control measures to prevent the emergence and spread of multidrug-resistant strains of *P. aeruginosa*.

The genotypic results for the rDNA 16S genes were positive to all isolates, allowing us to conclude that all of them were *P. aeruginosa*, as expected. The genes *bla*_KPC_, *bla*_CTX_, *bla*_SHV_, *bla*_Smp_, *bla*_TEM_, *bla*_OXA_, *bla*_Imp,_ *bla*_PER_, and *bla*_Vim_ were tested to verify genotypic resistance to β-lactams. In terms of genotypic resistance, the *bla*_KPC_ gene tested positive for all 6 isolates tested, while all the remaining genes tested negative. These results are in line with the assay carried out by Neyestanaki et al., where no *bla*_CTX_ or *bla*_Smp_ was detected in any of the isolates, although *bla*_KPC_ was also not detected [36]. Other assays have reported isolates carrying the *bla*_TEM_, *bla*_OXA_ and *bla*_PER_ genes, as well as isolates carrying the *bla*_Imp_ and *bla*_Vim_ genes [36,38]. Several studies have investigated the presence of resistance genes in *Pseudomonas aeruginosa* isolated from dogs, and some of them have reported the absence of certain genes. For example, one study found that the genes for metallo-β-lactamase enzymes (*bla*_Imp_, *bla*_Vim_, and *bla*_NDM_) were not detected in any of the *P. aeruginosa* isolates from dogs, indicating that these strains were unlikely to be resistant to carbapenem antibiotics [54]. These findings are encouraging and suggest that there may be *P. aeruginosa* strains in dogs that are not as resistant to antibiotics as some human isolates. However, it is important to note that resistance patterns can vary between bacterial strains and geographic regions, and that continuous monitoring of antibiotic resistance in veterinary medicine is crucial to ensuring the effective treatment of infections in dogs and to preventing the spread of antibiotic-resistant strains [55]. The establishment of the European Antimicrobial Resistance Surveillance Network for Veterinary Pathogens (EARS-Vet) in 2005 was a critical advancement in monitoring veterinary practices. The primary objective of this network is to create a uniform approach to AMR surveillance in veterinary pathogens throughout Europe. As such, the EARS-Vet initiative represents a significant step towards comprehending the prevalence and dissemination of AMR in veterinary medicine [56]. For the remaining resistance genes, no isolates were detected. In contrast, a study by Poonsuk et al. investigated the prevalence of antibiotic resistance genes in 60 *P. aeruginosa* strains isolated from canine and feline infections. None of the isolates were found to contain *aph*(3′)-IIb, *ant*(2”)-Ia and *aac*(6′)-IIb [57]. This study was in line with the results of our study.

All of the isolates showed the presence of the *opr*D gene for this porin, which is consistent with the findings of Haenni et al.’s study, where 11 out of 12 isolates had an amplified *opr*D gene and did not undergo any mutations [58]. The significance of investigating the *opr*D gene in *P. aeruginosa* among dogs is due to its role in producing a porin that governs the entry of antibiotics into bacterial cells. In the absence or mutation of this gene, the bacteria can become immune to specific antibiotics, resulting in complications in treating infections [59,60]. In dogs, *P. aeruginosa* infections can be particularly challenging to manage as they can cause a range of serious diseases, including skin infections, urinary tract infections, pneumonia, and sepsis [61]. The use of antibiotics is often necessary to treat these infections, but antibiotic-resistant *P. aeruginosa* is becoming increasingly common, making it essential to understand the mechanisms underlying this resistance. By studying the *opr*D gene in this bacterium, researchers can identify strains that are more likely to be resistant to antibiotics and develop more effective treatment strategies [59].

By studying the virulence genes in *P. aeruginosa*, researchers can identify the specific genes that are responsible for causing disease in dogs. This information can be used to identify risk factors for infection, improve infection control practices, and develop new therapies that target specific virulence factors. Investigating the virulence genes in *P. aeruginosa* is essential for comprehending the pathogenesis of infections in dogs and mitigating the possibility of infection and transmission to other animals and humans. All of the isolates in this study showed amplified virulence genes, including *exo*S, *exo*A, *exo*T, *exo*Y, *apr*A, *alg*D, and *plc*H (Figure 1). Moreover, 81% of the isolates also had the *tox*A virulence gene amplified. The studies conducted on canine ocular infection strains revealed that more than 81% of the isolates had the virulence gene *exo*S, while over 92% had the virulence gene exoY, and 96% had the virulence gene *exo*T [62]. Similarly, in other studies on the detection of genes *exo*S, *apr*A, *plc*H, *tox*A, and *las*B, all isolates showed amplification of *apr*A, *plc*H, and *las*B genes, while 87.5% of the isolates had the virulence gene *exo*S and 91.7% had the virulence gene *tox*A [55]. These findings are consistent with our results. The detection of these virulence genes in *P. aeruginosa* is not surprising given the bacterium’s well-established reputation for possessing a wide range of virulence factors. These factors play a critical role in its remarkable adaptability and infectious potential. The *exo*S, *exo*T and *exo*Y genes encode for an effector protein that inhibits the host’s immune response, while *exo*A encodes for the protein exotoxin A in order to cause tissue damage and inhibit protein synthesis in host cells [48,49]. The *apr*A gene encodes for the protein alkaline protease, and *plc*H gene encodes for the phospholipase C enzyme; these are involved in tissue degradation and can also contribute to the bacterium’s ability to evade the host immune response [44]. The *alg*D gene encodes for an enzyme that synthesizes a polysaccharide called alginate, an important component of the biofilm produced by *P. aeruginosa* [43]. The *plc*H gene encodes for the phospholipase C enzyme involved in the degradation of host cell membranes and can also contribute to the bacterium’s ability to evade the host immune response [43].

The *rhI*I gene, which encodes for a protein involved in the regulation of the bacteria’s iron uptake system, was present in a majority of the isolates. This was the case in approximately 81.5% of the isolates. The *rhIA/B* gene, which encodes for enzymes involved in the biosynthesis of the O-antigen of the lipopolysaccharide, was also highly prevalent, occurring in 92.6% of the isaoltes. The *pil*A gene, which encodes for the structural component of type IV pili, was present in a lower percentage of the isolates, with just a 40.7% presence rate. The absence of the virulence genes *las*R, *las*A, *rhI*R, and *pil*B suggests that the *P. aeruginosa* isolates from dogs in this study may not pose a significant risk of causing severe infections. *Las*R and *Las*A are known to be involved in quorum sensing and the production of virulence factors, respectively [43,44]. *Rh*IR is involved in the regulation of the bacteria’s response to iron starvation, while *Pil*B is involved in the assembly of type IV pili, which are important for bacterial motility and adhesion [43]. The absence of these genes in the isolates suggests that they may be less virulent and less able to colonize and infect their host. However, *P. aeruginosa* is known for having a highly mutatable genome, which means that it can undergo genetic changes that may result in the acquisition of new virulence mechanisms. This bacterial species has been shown to have a remarkable ability to adapt to different environments, including the host’s tissues, where it can cause infections [2]. Therefore, even though the results of this study suggest that the *P. aeruginosa* isolates from dogs may not pose a significant risk of causing severe infections due to the absence of certain virulence genes, it is important to note that these bacteria are highly adaptable and can acquire new mechanisms for infection [63]. Further studies will be necessary to investigate the molecular mechanisms involved in the pathogenesis of *P. aeruginosa* infections in dogs and to monitor the evolution of the bacteria in response to selective pressures. Based on a literature review, there seem to be limited data available on the prevalence of the virulence genes *las*R, *las*A, *rh*IR, and *pil*B in *P. aeruginosa* isolates specifically from dogs, highlighting the importance of our study. However, there are some studies that have investigated the prevalence of these genes in *P. aeruginosa* isolates from other sources, such as humans and the environment. A study conducted by Yumi Park and Sun Hoe Koo investigated the prevalence of carbapenem-resistant *P. aeruginosa* (CRPA) in patients with urinary tract infections (UTIs), and also examined the molecular characteristics and virulence factors of the isolated strains. In terms of the prevalence of virulence genes, the study found that the genes *las*R, *las*A, and *rhI*R were present in a significant proportion of the CRPA isolates. Specifically, the *las*R gene was detected in 96.7% of the isolates, the *las*A gene was detected in 93.3% of the isolates, and the *rh*IR gene was detected in 86.7% of the isolates [64]. Another study conducted by O’Connor et al. investigated the prevalence of virulence genes in 90 environmental *P. aeruginosa*. They authors found that the *las*R and *las*A genes were present in 50% and 56% of the isolates, respectively. The *rh*IR gene was present in all isolates, while the *pil*B gene was present in 92% of the isolates [65].

The microtiter assay is the most commonly used method for the analysis of biofilm biomass due to its accuracy and reproducibility [66,67]. Biofilm formation ability was considered as positive at a cut-off level of 0.240. We determined cut-off arbitrarily using the negative control (culture medium, 0.058) plus tree standard deviations (0.01). Levels of biofilm production were established based on the following classification criteria: weak biofilm formers: 0.240 < A570 ≤ 0.481 (2 × negative controls); moderate biofilm formers: 0.481 < A570 ≤ 0.962 (4 × negative controls); strong biofilm formers: A570 > 0.962. All isolates appeared to be weak biofilm producers. This produces an interesting result because in humans, the production of biofilms is typically strong. A study by Płókarz et al. investigated the prevalence of virulence factor genes and biofilm-forming ability in *P. aeruginosa* isolates from dogs and cats [68]. The study aimed to identify potential biomarkers to predict biofilm formation ability and guide treatment decisions. The study found that 90.6% of *P. aeruginosa* isolates from dogs and 86.4% from cats were capable of biofilm formation. The most prevalent virulence factor gene in both species was *ppy*R, followed by *psl*A, *fli*C, *nan*1, and *pel*A. Additionally, the presence of the *fli*C gene was significantly associated with biofilm-forming ability in dogs, while the absence of the *nan*1gene was significantly associated with biofilm-forming ability in cats. These findings suggest that the detection of specific virulence genes may serve as useful biomarkers for predicting biofilm-forming ability in *P. aeruginosa* infections in dogs and cats. This information could potentially guide treatment decisions and improve clinical outcomes for affected animals [68]. Another study by Pye et al. evaluated the biofilm-forming capacity of *P. aeruginosa* isolated from canine ears and its impact on antimicrobial susceptibility. The hypothesis was that biofilm-forming capacity is common among *P. aeruginosa* isolates causing otitis in dogs, and that biofilm-embedded bacteria would have a higher minimal inhibitory concentration (MIC) than planktonic bacteria [69]. The findings of the study revealed that 33 out of the total isolates, equivalent to 40%, were categorized as biofilm producers. Moreover, the biofilm MICs for all four antimicrobials were significantly higher than the MICs for planktonic bacteria (*p* < 0.05), suggesting that biofilm-embedded bacteria are more resistant to these drugs. The study’s conclusions suggest that biofilm production is common among *P. aeruginosa* isolates causing otitis in dogs and that it may play a role in the pathogenesis of the disease. The higher MICs for biofilm-embedded bacteria also suggest that treatment with neomycin, polymyxin B, gentamicin, or enrofloxacin may be less effective in treating chronic otitis caused by *P. aeruginosa*. Thereby, dogs can serve as a model for human infections; by studying *P. aeruginosa* biofilms in dogs, researchers can gain a better understanding of how these structures contribute to infection and develop more effective treatments [68,70].

## 4. Conclusions

This study investigated the resistance patterns of 27 clinical isolates of *P. aeruginosa* to various antimicrobial agents used in both human and veterinary medicine. The results showed varying degrees of resistance to different antibiotics, with resistance to β-lactam antimicrobials being the most common. The study also highlighted the importance of understanding *P. aeruginosa* resistance patterns in different populations and regions and emphasized the need for the responsible use of antibiotics and strict adherence to infection control measures to prevent the emergence and spread of multidrug-resistant strains of *P. aeruginosa*. This study’s results align with previous research from various global locations indicating that *P. aeruginosa* resistance patterns differ based on the population, geography, and antibiotic usage. Monitoring the antimicrobial resistance patterns of *P. aeruginosa* is crucial in order to minimize public health problems worldwide. Surveillance of this pathogen is necessary to prevent the emergence and spread of multidrug-resistant strains, which can pose significant challenges in clinical settings. By monitoring this pathogen, we can take proactive measures to protect public health and combat antimicrobial resistance.

## Figures and Tables

**Figure 1 vetsci-10-00343-f001:**
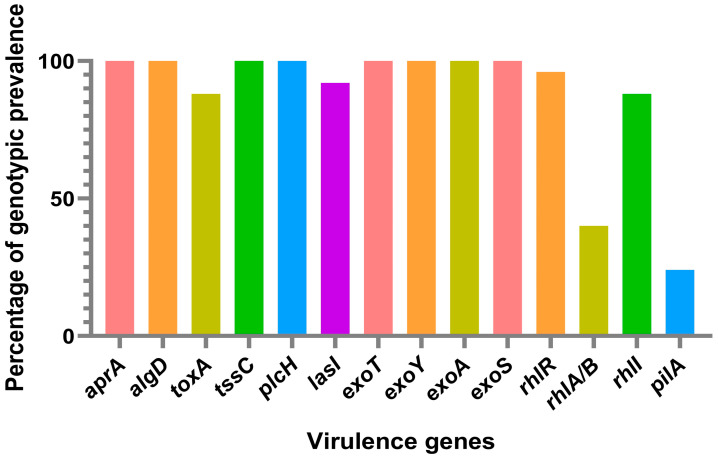
The percentage of each virulence gene found within the *P. aeruginosa* isolates derived from dogs at the INNO Veterinary Laboratory, Braga, Portugal, between November 2021 and December 2021.

**Table 1 vetsci-10-00343-t001:** Characteristics of the study population and collection sites of *P. aeruginosa* strains isolated from dogs.

Isolate	Sex	Age (Year)	Collection	Sample
D1	F	15	November	Chronic ulcerative Dermatitis
D2	M	12	November	Urine
D3	M		November	Ear exudate
D4	M	2	November	Ear exudate
D5	F	4 M	November	Skin exudate
D6	F	10	November	Ear exudate
D7	F	15	November	Ear exudate
D8	M	-	November	Skin exudate
D9	F	-	November	Ear exudate
D10	F	2	December	Vaginal exudate
D11	F	7	December	Skin exudate
D12	F	7	December	Urine
D13	M	-	December	Ear exudate
D14	M	8	December	Lip injury
D15	F		December	Ear exudate
D16	M	10	December	Ear exudate
D17	F	6	December	Ear exudate
D18	M	7	December	Ear exudate
D19	M	2	December	Ear exudate
D20	M	2	December	Ear exudate
D21	M	8	December	Ear exudate
D22	F	12	December	Ear exudate
D23	M	9	December	Ear exudate
D24	M	6	December	Ear exudate
D25	F	12	December	Ear exudate
D26	F	10	December	Ear exudate
D27	F	8	December	Ear exudate

M—Male; F—Female.

**Table 2 vetsci-10-00343-t002:** Primer sequences for PCR used to amplify the different genes.

Name	Sequence (5′ → 3′)	Length (bp)	References
*bla* _TEM_	F: ATTCTTGAAGACGAAAGGGC R: ACGCTCAGTGGAACGAAAAC	1150	[31]
*bla* _SHV_	F: CACTCAAGGATGTATTGTG R: TTAGCGTTGCCAGTGCTCG	885	[32]
*bla* _CTX_	F: CGATGTGCAGTACCAGTAA R: TTAGTGACCAGAATCAGCGG	585	[33]
*bla* _PER_	F: ATGAATGTCATTATAAAAGC R: AATTTGGGCTTAGGGCAGAA	920	[34]
*bla* _SME_	F: ACTTTGATGGGAGGATTGGC R: ACGAATTCGAGCATCACCAG	551	[35]
*bla* _KPC_	F: GTATCGCCGTCTAGTTCTGC R: GGTCGTGTTTCCCTTTAGCC	638	[36]
*bla_IMP_*	F: GTTTATGTTCATACTCG R: GGTTTAAAAAACAACCAC	432	[36]
*bla* _Smp_	F: AAAATCTGGGTACGCAAACG R: ACATTATCCGCTGGAACAGG	271	[37]
*bla* _Vim_	F: TTTGGTCGCATATCGCAACG R: CCATTCAGCCAGATCGGCAT	500	[38]
*bla* _Vim-2_	F: AAAGTTATGCCGCACTCACC R: TGCAACTTCATGTTATGCCG	815	[39]
*bla* _NDM_	F: GGTTTGGCGATCTGGTTTTC R: CGGAATGGCTCATCACGATC	621	[36]
*bla* _OXA_	F: CCAAAGACGTGGATG R: GTTAAATTCGACCCCAAGTT	813	[32]
*aac*(6′)-Ie-*aph*(2″)-Ia	F: CCAAGAGCAATAAGGGCATA R: CACTATCATAACCACTACCG	220	[40]
*aph*(3′)-IIIa	F: GCCGATGTGGATTGCGAAAA R: GCTTGATCCCCAGTAAGTCA	292	[40]
aac(3)-II	F: ACTGTGATGGGATACGCGTC R: CTCCGTCAGCGTTTCAGCTA	237	[41]
aac(3)-III	F: CACAAGAACGTGGTCCGCTA R: AACAGGTAAGCATCCGCATC	195	[41]
aac(3)-IV	F: CTTCAGGATGGCAAGTTGGT R: TACTCTCGTTCTCCGCTCAT	286	[40]
*ant*(4′)-Ia	F: GCAAGGACCGACAACATTTC R: TGGCACAGATGGTCATAACC	165	[40]
*ant*(2′)-I	F: ATGTTACGCAGCAGGGCAGTCG R: CGTCAGATCAATATCATCGTGC	188	[41]
*oprD*	F: TCCGCAGGTAGCACTCAGTTC R: AAGCCGGATTCATAGGTGGTG	191	[42]
*pil*B	F: TCGAACTGATGATCGTGG R: CTTTCGGAGTGAACATCG	408	[43]
*pil*A	F: ACAGCATCCAACTGAGCG R: TTGACTTCCTCCAGGCTG	1675	[43]
*apr*A	F: ACCCTGTCCTATTCGTTCC R: GATTGCAGCGACAACTTGG	140	[44]
*tox*A	F: GGTAACCACGTCAGCCACAT R: TGATGTCCAGGTCATGCTTC	352	[45]
*tss*C	F: CTCCAACGACGCGATCAAGT R: TCGGTGTTGTTGACCAGGTA	150	[46]
*plc*H	F: GCACGTGGTCATCCTGATGC R: TCCGTAGGCGTCGACGTAC	608	[43]
*las*A	F: GCAGCACAAAAGATCCC R: GAAATGCAGGTGCGGTC	1075	[43]
*las*B	F: GGAATGAACGAAGCGTTCTC R: GGTCCAGTAGTAGCGGTTGG	284	[43]
*las*R	F: CGGGTATCGTACTAGGTGCATCA R: GACGGGAAAGCCAGGAAACTT	1100	[44]
*lasI*	F: ATGATCGTACAAATTGGTCGGC R: GTCATGAAACCGCCAGTCG	605	[47]
*exo*U	F: ATGCATATCCAATCGTTG R: TCATGTGAACTCCTTATT	2000	[44]
*exo*S	F: CGTCGTGTTCAAGCAGATGGTGCTG R: CCGAACCGCTTCACCAGGC	444	[48]
*exo*A	F: GACAACGCCCTCAGCATCACCAGC R: CGCTGGCCCATTCGCTCCAGCGCT	396	[49]
*exo*Y	F: CGGATTCTATGGCAGGGAGG R: GCCCTTGATGCACTCGACCA	289	[49]
*exo*T	F: AATCGCCGTCCAACTGCATGCG R: TGTTCGCCGAGGTACTGCTC	159	[49]
*rhl*R	F: CAATGAGGAATGACGGAGGC R: GCTTCAGATGAGGCCCAGC	730	[47]
*rhl*I	F: CTTGGTCATGATCGAATTGCTC R: ACGGCTGACGACCTCACAC	625	[47]
*rhl*A/B	F: TCATGGAATTGTCACAACCGC R: ATACGGCAAAATCATGGCAAC	151	[50]
*alg*D	F: CGTCTGCCGCGAGATCGGCT R: GACCTCGACGGTCTTGCGGA	313	[43]

**Table 3 vetsci-10-00343-t003:** Antimicrobial resistance phenotypes of *Pseudomonas aeruginosa* isolated from dogs (*n* = 27).

Class and⁄or Antimicrobial	Breakpoints (mm; S≥/R<)	Dogs (*n* = 27)	
		Number of Resistant Strains	Percentage of Resistant Strains (%)
**β-Lactams**			
Cefepime	50/21		0
Ceftiofur	**	16	59
Cefovecin	**	20	74
Aztreonam	50/18		0
Ceftazidime *	18/15 *	8	30
Doripenem	50/22		0
Imipenem	50/20	8	30
Meropenem	20/14	1	4
Ticarcillin-clavulanic acid	50/18	1	4
**Fluoroquinolones**			
Ciprofloxacin	50/26	2	7
Enrofloxacin	**	2	7
Marbofloxacin	**	1	4
**Aminoglycosides**			
Amikacin	15/15		0
Tobramicyn	18/18		0
Gentamicin	15/15	2	7

* CLSI 2021; ** Minimum Inhibitory Concentrations by VITEK 2^®^ COMPACT.

## Data Availability

Not applicable.

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
