# Peer review of "The Impact of the Virulence of Pseudomonas aeruginosa Isolated from Dogs"

_vetsci, 2023, doi:10.3390/vetsci10050343_

Round 1

Reviewer 1 Report

The paper is well written and it is of interest for the veterinary and scientific community.

Author Response

The authors thank all reviewers for their constructive comments, which allowed significant improvement of the manuscript. We proceeded with the revision of the manuscript in the light of the comments received and brief responses to the reviewers’ comments are included. All the modifications in the text are marked using “Track Changes” function.

Reviewer 2 Report

General comments:

This article deals with the very important issue of antimicrobial resistance of P. aeruginosa. In fact, P. aeruginosa is considered by WHO a first priority microorganism regarding the problem of antibiotic tolerance. The bacterium has an opportunistic nature and the infection usually occurs when there is some short of breach of the normal barriers (i.e., skin). Susceptible human population are those with implanted medical devices or weakened immune system. This study presents the results found in a sample of 27 dogs diagnosed with P. aeruginosa infection along with a wealth of information from other similar studies. The object of the article is within the scope of the journal "Veterinary Sciences", having both veterinary and One Health aspects.

Specific comments:

L 29 The study aimed: This study aimed

L 34 All isolates were resistant to β-lactam antibiotics: remove, already mentioned/ unless by isolates you mean a subgroup of the tested samples.

L 38, 39 Study compared P. aeruginosa resistance patterns worldwide: improve grammar. Generally minor grammar issues are found throughout the text.

L 51 produce several virulence factors, including toxins: instead of “produce” better release or excrete

L 68 produces a variety of toxins that contribute to its virulence [8]: as in previous comment

L 77 P. aeruginosa is a bacterial pathogen that can cause infections in dogs: “is a bacterial pathogen” is repetition, has been already mentioned. Pls remove.

L 80. Please provide better description of how the infection occurs. Usually, the infection occurs by environmental exposure and requires a breached barrier (i.e., skin) or inoculation at a favorable niche (i.e., ear canal).

L 95 ,96 Understanding the mechanisms of resistance and the most effective treatment options can help veterinarians provide better care for dogs with P. aeruginosa infections [23, 24]: Understanding the mechanisms of resistance can help veterinarians decide the most effective treatment options for dogs with P. aeruginosa infections [23, 24]

L 97 Another point is that…”: pls fix grammar use correct scientific-manuscript-style of words

L 99,100 “Thus, understanding the transmission and prevention of P. aeruginosa infections in dogs can have implications for public health [25].” Improve the sentence. Say i.e., “Thus, understanding the transmission and prevention of P. aeruginosa infections in dogs can have implications for public health as has been suggested by others [25].”

L 111 Instead of “samples” use “isolates”.

L 111 Describe the residence of the dogs participating in the study. INNO Laboratory location also.

L 184 Results and discussion. In general, you should present if possible similar data from other studies on P. aeruginosa to dogs or humans from the same area and examine how they compare. The same area could be i.e., the country or the adjacent countries. This is relevant as you correctly suggest potential geographical patterns.

L 207 fix grammar.

L 210 “In America”: please specify.

L 212 The sentence describes findings “in America” but the reference [52] is about Iran. Please cite the primary article.

L 217-219 repetition of what has been already mentioned in the previous sentence. Pls remove.

L 381-385 these sentences are somewhat repetitive, the same meaning could be conveyed by less text. Also, a full stop to the last line is needed.

Author Response

(The authors gave the same response as above.)

Reviewer 3 Report

Main concern: there are 76  isolates from 29 dogs but it is not specified how many isolates from the same individual and this infers prevalence and statistical  analysis.  In my opinion you should make the study with a larger number of dogs increasing the collection period. It is not clear if some isolates carried multiple resistance genes.  Why did you perform PCR of resistance genes based on fhenotypic results considering that sometimes genes are present but not expressed until specific circumstances.? In fact Bla kpc that you detected in your isolates  can give also carbapanem resistance  in human clinical isolates of P. aeruginosa for example.  In comparing the results to other studies it is very important to consider the different technical methods used because these can affect the results although regional, therapeutic and life styles difference are also relevant .The majority  of isolates are from ear exudate  were dogs affected by otitis? What is the therapy in these cases in dogs in your region?

Same concepts are repeated in different sentences in the introduction and in discussion as well. 

Author Response

(The authors gave the same response as above.)

Reviewer 4 Report

The manuscript entitled “The impact of the virulence of Pseudomonas aeruginosa isolated from dogs: A Public Health Problem” is interesting, relevant, and well written and should be included in the special issue “One Health Challenges and Opportunities — Animals, Humans and Their Interconnected Ecosystems”.

The study was well conducted, the results are adequately treated, and are presented and discussed in a clear, logical, and coherent way.

Therefore, the manuscript should be considered for publication after some minor corrections/revisions are made.

Author Response

(The authors gave the same response as above.)

Round 2

Reviewer 3 Report

What is the therapy in these cases in dogs in your region? Sorry, probably I was not very clear but I was wondering if you have an idea on the main antibiotic prescribed or used by vets  in Portugal  in otitis cases in dogs not the specific treatment for each animal.  These information are usually available among vets and this can justify the prevalence of one antibiotic resistance compared to other one.

In Material and methods PCR conditions are not described ( programs and cycles?) . Single PCR only or Multiplex?  Do you follow the methods described in the references?

(Liofilchem, Rosetodegli, Abruzzi, Italy) medium at 37°C (lane 124) The place is  Roseto degli Abruzzi 

Author Response

Thank you very much for taking the time to review and provide corrections to our work, We truly appreciate your valuable feedback.
